# Flexible Real-Time Polymerase Chain Reaction-Based Platforms for Detecting Deafness Mutations in Koreans: A Proposed Guideline for the Etiologic Diagnosis of Auditory Neuropathy Spectrum Disorder

**DOI:** 10.3390/diagnostics10090672

**Published:** 2020-09-04

**Authors:** Sang-Yeon Lee, Doo-Yi Oh, Jin Hee Han, Min Young Kim, Bonggi Kim, Bong Jik Kim, Jae-Jin Song, Ja-Won Koo, Jun Ho Lee, Seung Ha Oh, Byung Yoon Choi

**Affiliations:** 1Department of Otorhinolaryngology-Head and Neck Surgery, Seoul National University Bundang Hospital, Seoul National University College of Medicine, Seongnam 13620, Korea; maru4843@hanmail.net (S.-Y.L.); dooyi9@gmail.com (D.-Y.O.); flyswan@daum.net (J.H.H.); mn911@hanmail.net (M.Y.K.); jjsong96@gmail.com (J.-J.S.); jwkoo99@snu.ac.kr (J.-W.K.); 2Department of Otorhinolaryngology-Head and Neck Surgery, Chungnam National University College of Medicine, Daejeon 35015, Korea; elprimero_@naver.com (B.K.); cellokimbj@gmail.com (B.J.K.); 3Department of Otorhinolaryngology-Head and Neck Surgery, Seoul National University Hospital, Seoul National University College of Medicine, Seoul 03080, Korea; junlee@snu.ac.kr (J.H.L.); shaoh@snu.ac.kr (S.H.O.); 4Sensory Organ Research Institute, Seoul National University Medical Research Center, Seoul 03080, Korea

**Keywords:** U-TOP™ HL Genotyping Kit Ver2, hearing loss, auditory neuropathy spectrum disorder

## Abstract

Routine application of next-generation sequencing in clinical settings is often limited by time- and cost-prohibitive complex filtering steps. Despite the previously introduced genotyping kit that allows screening of the 11 major recurring variants of sensorineural hearing loss (SNHL) genes in the Korean population, the demand for phenotype- and variant-specific screening kits still remains. Herein, we developed a new real-time PCR-based kit (U-TOP™ HL Genotyping Kit Ver2), comprising six variants from two auditory neuropathy spectrum disorder (ANSD) genes (*OTOF* and *ATP1A3*) and five variants from three SNHL genes (*MPZL2*, *COCH*, and *TMC1*), with a distinct auditory phenotype, making this the first genotyping kit dedicated to ANSD. The concordance rate with Sanger sequencing, sensitivity, and specificity of this genotyping kit were all 100%, suggesting reliability. The kit not only allows timely and cost-effective identification of recurring *OTOF* variants, but it also allows timely detection of cochlear nerve deficiency for those without *OTOF* variants. Herein, we provide a clinical guideline for an efficient, rapid, and cost-effective etiologic diagnosis of prelingual ANSD. Our study provides a good example of continuing to update new key genetic variants, which will continuously be revealed through NGS, as targets for the newly developed genotyping kit.

## 1. Introduction

The realization of the importance of genomics in biological sciences has led to the use of genomics information for the development of precision medicine [1]. Sensorineural hearing loss (SNHL) is one of the most common neurosensory disorders in humans, affecting approximately 1 in every 1000 newborns and 1 in every 300 children by 4 years of age [2]. The early detection of SNHL and timely intervention are necessary to avoid devastating disabilities that are associated with auditory deprivation during the most sensitive period of brain development and plasticity [3,4]. The identification of the genetic etiology of SNHL could allow for the prediction of hearing prognosis and facilitate the establishment of an appropriate auditory rehabilitation strategy on the basis of underlying pathophysiological mechanisms [5,6]. Several studies have shown that congenital deafness is largely associated with genetic etiology, with a frequency of 50% or more [7]. Moreover, existing evidence indicates that there is a meaningful genetic contribution of Mendelian inheritance to post-lingual sensorineural deafness [8]. Recently, next-generation sequencing (NGS), based on molecular genetic testing, has been widely employed to identify candidate variants responsible for SNHL [9]. When time and cost is not an issue (i.e., for research purposes), and especially when blind to a specific auditory phenotype or genotype–phenotype correlation, NGS would be an extremely powerful tool that can scan all deafness genes. However, routine application of NGS in the everyday clinical setting is often limited by time and cost of the highly complex filtering steps [10,11,12]. As such, the development of molecular genetic testing that is more time- and cost-effective, with rapid screening capability, may be necessary in the context of precision medicine for SNHL.

We previously developed a genetic diagnostic kit that enabled screening of the prevalent variants of five major prelingual deafness genes in the Korean population [13]. The kit was highly flexible, time efficient, and cost effective, ideal for the everyday clinical setting [13]. Nevertheless, given an extreme etiologic heterogeneity, there is still demand for the development of additional convenient screening kits that ensure coverage of other important deafness genes. This is specifically the case when there is either a characteristic auditory phenotype or highly prevalent variants in a gene. For example, we recently found that four prevalent *OTOF* variants account for about 70% of Korean prelingual auditory neuropathy spectrum disorder (ANSD) with the anatomically intact cochlear nerve [14], making it unnecessary to go through other deafness genes irrelevant to such a phenotype. Furthermore, the predominant variant, p.Arg1939Gln, of *OTOF*, was frequently missed during NGS-based targeted exome sequencing due to capture failures and the poor coverage of the last exon of the cochlear isoforms of *OTOF* [14,15]. Considering this, a genotyping kit that facilitates screening exclusively for such prevalent *OTOF* variants would minimize the time and cost of NGS in Korean prelingual ANSD. This issue implicates some major variants of other deafness genes as well. Indeed, several prevalent variants that exert a potential hotspot/founder effect or manifest specific auditory phenotypes have been recently identified in the Korean population with varying degrees of SNHL [16,17,18,19,20].

Encouraged by the rising demand for phenotype-specific and variant-specific screening kits for deafness, we developed a genetic diagnostic kit (U-TOP™ HL Genotyping Kit Ver2), characterized by real-time PCR-based melting array techniques and peptide nucleic acid (PNA) probes, for screening of 11 variants from five deafness genes, which comprised two ANSD genes (*OTOF* and *ATP1A3*) and three SNHL genes (*MPZL2*, *COCH*, and *TMC1*), with a distinct auditory phenotype among Koreans. Any laboratory equipped with a real-time PCR machine can easily use this kit, without the limitation that at least one of the sequencers for Sanger sequencing or NGS must exist. In this study, we present the efficacy of this kit for detection of the corresponding variants from our previously established cohort in comparison with Sanger sequencing. We also prospectively tested the diagnostic yield of this kit among a prospectively recruited prelingual ANSD cohort. Based on this, we herein provide a clinical guideline on how otologists/pediatricians should deal with prelingual ANSD in children under this era of precise medicine and customized auditory rehabilitation.

## 2. Materials and Methods

### 2.1. Ethics Statement

This study was approved by the institutional review boards (IRBs) of Seoul National University Bundang Hospital (IRB-E-1905/540-001, 17 July 2020). We obtained written informed consent from all participants in this study. For child participants, written informed consent was obtained from their parents or guardians. We confirmed that all methods were carried out in accordance with relevant guidelines and regulations.

### 2.2. Participants

We collected patient samples with SNHL and family members who visited the clinics of SNUBH from May 2015 through May 2018, as well as those who executed genetic testing for causative variants. Among the 121 participants, we obtained 72 positive samples that had at least one of the 11 variants from the five genes (*OTOF*, *ATP1A3, MPZL2*, *COCH*, and *TMC1*) in homozygous, heterozygous, or compound heterozygous, as confirmed by Sanger sequencing. Forty-nine normal control subjects without any of the 11 variants were also collected.

### 2.3. Validation of the Real-Time PCR-Based Melting Array Genetic Diagnostic Kit Ver2

To validate the performance of the molecular diagnostic kit, which is referred to as the U-TOP™ HL Genotyping Kit Ver2 (SeaSun Biomaterials) [13], we tested the DNA samples and compared the results obtained using the U-TOP™ HL Genotyping Kit Ver2 with those obtained by Sanger sequencing. All experiments with randomly anonymous samples were blinded in terms of previously identified variants. Genomic DNA (gDNA) was extracted from the whole blood samples by using standard protocols (Gentra Puregene Blood Kit, Catalog No. 158389; Qiagen, Venlo, The Netherlands). The gDNA samples had 260 nm/280 nm absorbance ratios of over 1.5.

### 2.4. Real-Time PCR

Real-time PCR was performed using the U-TOP™ HL Genotyping Kit Ver2 with a CFX96 Real-Time PCR Detection System (Bio-Rad, Hercules, CA, USA). Here, 11 variants from five deafness genes, comprised of two ANSD genes (*OTOF* and *ATP1A3*) and three SNHL genes (*MPZL2*, *COCH*, and *TMC1*), were examined by using this kit as per the manufacturer’s manual [13]. The data were analyzed with Bio-Rad CFX manager v1.6 software (Bio-Rad). These variants were characterized by the fluorescence signal of the detection probes and corresponding to T_m_, according to the standard protocol outlined by the manufacturer′s manual [13].

### 2.5. Sanger Sequencing

All gDNA samples were examined individually with each of the 8 sets of primers in both directions and were confirmed by Sanger sequencing (Macrogen Inc., Seoul, Korea).

### 2.6. Statistical Analysis

To evaluate the performance of the U-TOP™ HL Genotyping Kit Ver2, the sensitivity, specificity, and 95% confidence intervals (CI) were calculated as described previously [13]. To assess the accuracy between the U-TOP™ HL Genotyping Kit Ver2 and Sanger sequencing, we calculated the kappa (κ) statistic, which reflects perfect agreement when the κ value is 0.81 and 1.00 [13]. Cohen′s Kappa value between the two tests was determined as a reference to the following equation.
*n* = (*Z_α/2_* + *Z_β_*)^2^*p*(1 − *p*)/(*δ* − |*p* − *p*_0_|)^2^
where *p* = the average positive predictive value (PPV) and negative predictive value (NPV) values from the reference articles; *p*_0_ = the expected PPV and NPV for this study (equivalence); *Z*_α/2_ = 1.96; *Z_β_* = 0.842.

## 3. Results

### 3.1. Design and Establishment of the U-TOP™ HL Genotyping Kit Ver2 

Based on previous Korean and East Asian reports from several leading institutes, including ours, we selected 11 variants from five genes (*ATP1A3, COCH, OTOF, MPZL2*, and *TMC1*), which were frequently detected and were previously reported to cause distinctive audiological phenotypes. In detail, *OTOF* variants (p.Glu841Lys, p.Arg1856Trp, p.Leu1011Pro, p.Tyr1064Ter, and p.Arg1939Gln) were highly prevalent among Korean and East Asian prelingual ANSD (Appendix A). *OTOF* variants, including a founder variant (p.Arg1939Gln) among Koreans, account for approximately 90% of Korean prelingual ANSD cases with anatomically intact cochlear nerves [14]. Consistent with this, *OTOF* p.Arg1939Gln was found in 20 of the 26 alleles (76.9%) among Japanese congenital or early-onset ANSD cases and the founder effect was determined for this variant [21]. Additionally, the *ATP1A3* Glu818Lys variant can cause postlingual-onset auditory synaptopathy, which is frequently accompanied by a distinct CAPOS (cerebellar ataxia, areflexia, pes cavus, optic atrophy, and sensorineural hearing loss) syndrome [16]. Although CAPOS syndrome is a rare disease that has been reported in less than 30 subjects thus far, the *ATP1A3* variant, p.Glu818Lys, was detected in all cases [22]. In collaboration with our study [16], a recent study also demonstrated evidence for ANSD in most subjects with CAPOS syndrome caused by *ATP1A3* p.Glu818Lys [23]. Furthermore, this kit would help to screen potential hotspot/founder variants that manifest specific auditory phenotypes. Very recently, we confirmed that a high proportion of Mendelian genetic contribution was due particularly to the p.Gln74* *MPZL2* variant amongst patients with pediatric-onset mild-to-moderate SNHL [18]. The *COCH* variants are suggested to be a frequent cause of progressive cochleovestibular dysfunction in Koreans eventually requiring cochlear implantation [17]. Specifically, we reported two distinct *COCH* variants (p.Gly38Asp and p.Cys162Tyr), demonstrating a potential genotype-cochleovestibular phenotype correlation [17]. According to our recent work, the genetic etiology has been reported in 21 (52.5%) of 40 postlingually deafened cochlear implantees. Among them, the most frequent causative etiology was the *TMC1* missense dominant variant, p.Asp572Asn, which was detected from three cases. Similarly, the *TMC1* variant (p.Asp572Asn) accounts for about 4.4% (3/68) of progressive, postlingual autosomal dominant nonsyndromic hearing loss (ADNSHL) in the Chinese population [24]. In addition, the *TMC1* variant (p.Arg34*) should be tested for in the genetic evaluation of subjects with autosomal recessive nonsyndromic hearing loss (ARNSHL) from North African and Southwest Asia [25].

A total of 11 variants were divided into three sets and were tested for validation of the efficacy of the U-TOP™ HL Genotyping Kit Ver2. Set 1 comprised three *OTOF* variants (p.Glu841Lys, p.Arg1856Trp, and p.Leu1011Pro) and one *TMC1* variant (p.Asp572Asn). Set 2 comprised two *COCH* variants (p.Gly38Asp and p.Cys162Tyr), one *OTOF* variant (p.Tyr1064Ter), and one *TMC1* variant (p.Arg34*). Set 3 comprised one *OTOF* variant (p.Arg1939Gln), one *MPZL2* variant (p.Gln74*), and one *ATP1A3* variant (p.Glu818Lys). Individually, all variants showed distinct melting curves that corresponded to each allele with a specific melting temperature (T_m_) value for each diagnostic melting peak. The representative melting peaks that corresponded to both the heterozygous state and, whenever available, the homozygous state of each variant, as well as to those of the wild-type sequence, are depicted in Figure 1. While the heterozygous variants usually exhibited dual melting peaks at T_m_, except for two variants as a single heterozygous state (p.Arg1939Gln of *OTOF* and p.Arg34* of *TMC1*), the homozygous variant and wild-type sequence consistently exhibited a single melting peak at T_m_.

### 3.2. Characteristics of Samples for Validation of the Efficacy of the U-TOP™ HL Genotyping Kit Ver2 

A total of 121 samples (72 samples with variants and 49 negative control samples) were prepared from our previously established cohort for validation of the detection efficacy of the newly developed genotyping kit for the target variants (Appendix A). Among the 121 samples, 72 samples were positive for carrying at least one of the 11 target variants as a single heterozygous (*n* = 45 (62.5%)), compound heterozygous (*n* = 18 (25.0%)), or homozygous (*n* = 9 (12.5%)) state (Figure 2A). In total, 99 alleles from 72 samples carried at least one target variant and 15 different genotypes were observed (Figure 2B). The proportion of genetic load of *OTOF, COCH, TMC1, MPZL2*, and *ATP1A3* among the 99 alleles was 55.6% (55/99), 14.1% (14/99), 13.1% (13/99), 9.1% (9/99), and 8 (8.1%), respectively. When it comes to the frequency of each variant, p.Arg1939Gln of *OTOF* was the most common, with an allele frequency of 22.1% (21/99, 13 single heterozygotes and 4 homozygotes), followed by the variant Glu841Lys of *OTOF* as the second most common (12/99, 12.1%) (Figure 2C).

### 3.3. Diagnostic Sensitivity, Specificity, and Concordance Rate

As compared with Sanger sequencing, the clinical sensitivity and specificity of the newly developed genotyping kit for the detection of target variants were 100% (95% confidence interval (CI): 94.9–100.0%) and 100% (95% CI: 92.7–100.0%), respectively (Table 1). A comparison study that included 127 clinical samples showed a concordance rate of 100% (95% CI: 96.9–100.0%) between the two methods. Accordingly, these data strongly indicate that the results obtained using U-TOP™ HL Genotyping Kit Ver2 were in perfect agreement with those obtained using Sanger sequencing.

### 3.4. Clinical Implications of the U-TOP™ HL Genotyping Kit Ver2 for ANSD

We also determined the diagnostic yield of the U-TOP™ HL Genotyping Kit Ver2 in the molecular diagnosis of prelingual ANSD by investigating six, prospectively recruited, consecutive cases with prelingual ANSD (Table 2). All six subjects manifested bilateral ANSD, as defined by the electrophysiological tests that exhibited “absent ABR,” but with the preservation of otoacoustic emissions or cochlear microphonics (Appendix A). Importantly, the U-TOP™ HL Genotyping Kit Ver2 detected the homozygous missense variant p.Arg1939Gln of *OTOF* from two of the seven probands (33.3%), making it unnecessary to get the internal auditory canal (IAC) magnetic resonance imaging (MRI) from these two subjects. Among the other four probands who do not carry *OTOF* variants, two (50.0%) turned out to have a bilateral cochlear nerve deficiency (CND) based on IAC-MRI. In one proband (SB515-981), a meticulous review of the medical histories revealed predisposing factors associated with ANSD, such as prematurity and hypoxic damage. Interestingly, the hearing of one proband (SB554-1029) in whom neither the genotyping kit nor IAC-MRI revealed any abnormality, showed improvement over time. These results suggest that the newly developed genotyping kit not only allows the timely and cost-effective identification of *OTOF* variants prevalent in Korean prelingual ANSD subjects, but it also significantly contributes to timely imaging tests that allow the detection of cochlear nerve deficiency for those lacking these target *OTOF* variants. Indeed, it is worth noting that the correct etiologic diagnosis of prelingual ANSD of these six subjects were possible within one month (median 20 days, range 11–52 days), which would not have been possible without the application of this new genotyping kit.

## 4. Discussion

This newly developed genotyping kit (U-TOP™ HL Genotyping Kit Ver2), which is a real-time PCR-based method using the melting array techniques and PNA probes, merits special attention, because this—to the best of our knowledge—is the first genotyping kit highly specialized to diagnose ANSD, of which the molecular etiology has remained enigmatic even in this NGS era. Five prevalent variants from *OTOF,* and one recurring variant from *ATP1A3*, which are causative for prelingual ANSD and peri/postlingual ANSD, respectively, were included in this kit. Additionally, five recurring variants from three genes (*MPZL2*, *COCH*, and *TMC1*) that are known to cause a distinct auditory phenotype among Koreans were also included to complement the previously introduced U-TOP™ HL Genotyping Kit Ver1 in terms of coverage of recurring variants.

The analytical performance of the newly developed genotyping kit was evaluated by comparing it with Sanger sequencing in 121 clinical samples with varying degrees of SNHL. The newly developed genotyping kit showed a concordance rate of 100% with Sanger sequencing. The sensitivity and specificity of the genotyping kit were 100% with a Kappa value of 1.00, strongly indicating that the newly developed kit is reliable for screening SNHL in the Korean population.

Recent studies have shown a higher genetic load of *OTOF* variants explaining 91% of prelingual ANSD in Koreans with an anatomically intact cochlear nerve [14]. Furthermore, understanding the distribution pattern of the Korean *OTOF* alleles, which are predominantly concentrated on certain exons among Korean prelingual ANSD, could provide a justification for proposing a set of the most frequent *OTOF* variants to be included in the first-line screening [14,15]. As a result, the new genotyping kit developed in this study exclusively comprises the five most prevalent missense variants of *OTOF* causing prelingual ANSD in the Korean population, showing perfect agreement with Sanger sequencing analysis. Recently, the authors disclosed that the NGS-based targeted resequencing of the known 134 deafness genes does not necessarily guarantee the accurate detection of causative variants, particularly, the major allele (p.Arg1939Gln) of Korean ANSD with prelingual onset, due to capture failures and poor coverage of the last exon of the cochlear isoforms of *OTOF* [14,15]. Furthermore, Sanger sequencing of *OTOF* is often hindered by the large size of this gene, which comprises 46 exons, and additionally, not all clinics are equipped with a sequencer for Sanger sequencing [26]. Another de novo, recurring variant p.Glu818Lys in *ATP1A3*, which causes enigmatic perilingual/postlingual ANSD and is sometimes accompanied by CAPOS syndrome [16], was also included in this kit, making this kit an exceptionally efficient and accurate screening tool for Korean ANSD children. The beauty of this kit also lies in the fact that all the reactions can be achieved within a couple of hours.

Given this, early molecular etiologic diagnosis of ANSD children offers two advantages. First, early acquisition of genetic information using the genotyping kit allows the avoidance of unnecessary imaging studies for those with target *OTOF* variants, as evidenced by our two probands (SB543-1015 and SB) in the present study. Second, and conversely, the use of this kit can provide a guide for timely imaging to reveal ANSD-related anatomical abnormalities for those who lack these target *OTOF* variants. Previous studies consistently reported a relatively high incidence of abnormal brain MRI findings among children with ANSD [27,28]. For example, Roche et al. demonstrated one or more abnormal MRI findings in nearly 65% of children with ANSD, suggesting that MRI is the initial imaging technique of choice for children with ANSD [27]. Specifically, the presence of CND is largely associated with prelingual ANSD [29,30], particularly with the characteristics of unilateral auditory neuropathy [30]. Our prospective data showed that two (50.0%) of the four probands who do not carry *OTOF* variants based on the genotyping kit had bilateral CND based on IAC-MRI. Previous studies have also shown that CND is significantly associated with poor speech intelligibility or perception after cochlear implantation, posing potentially limited effectiveness and uncertain cost-benefits [31,32]. In addition, previous studies suggested that the sensitive period for successful CI outcomes for children with *OTOF*-related DFNB9 may be narrower than that for those with *SLC26A4*- and *GJB2*-related deafness [14,33]. Supporting this, abnormal cortical auditory evoked potentials and delayed postsynaptic neurotransmission in children with ANSD may reflect abnormal maturation of the central auditory system, necessitating early intervention [34,35]. Collectively, *OTOF*-related DFNB9 with early intervention, i.e., before the age of 24 months, is considered a good candidate for CI with a good prognosis for outcome. Indeed, the application of this genotyping kit prompted the completion of etiologic diagnosis of most prelingual ANSD within approximately one month in our present study, which suggests that U-TOP™ HL Genotyping Kit Ver2 would significantly contribute to timely intervention and precise auditory rehabilitation of children with ANSD in Korea. Based on what we observed, as depicted in Figure 3, we propose a diagnostic pipeline for a more efficient, rapid, and cost-effective etiologic diagnosis of prelingual ANSD and provide a clinical guideline on how otologists/pediatricians should deal with prelingual ANSD children in this era of precise medicine and customized auditory rehabilitation. However, the relatively small sample size inherent from a pilot study precludes us from labeling the diagnostic yield of this kit for genetically diagnosing prelingual ANSD as 33%. A longitudinal follow-up study with a large cohort should be warranted to validate our preliminary results.

The implementation of the U-TOP™ HL Genotyping Kit Ver2 is not necessarily limited to ANSD. Indeed, the development of this new kit was based on the recent identification of several prevalent variants exerting a potential hotspot/founder effect or manifest specific auditory phenotypes in the Korean hearing-impaired population [15,16,17,18,19,20]. This new kit was designed to detect as many recurring and important deafness-causing variants as possible in conjunction with the previous version, U-TOP™ HL Genotyping Kit Ver1. By implementing this new kit, we expect that an additional 10.8% of the Korean pediatric-onset mild-to-moderate SNHL can be explained by the p.Gln74* *MPZL2* variant, when compared with using only the previous kit. In addition, we expect that this new kit would help to predict the clinical course of DFNA9 subjects. Previously, our group reported distinct vestibular phenotypes depending on the location of *COCH* variants in DFNA9 subjects, indicating a potential genotype–phenotype correlation [17]. In detail, the p.Gly38Asp variant is closely linked to bilateral vestibulopathy, while the p.Cys162Try variant is likely to manifest a Meniere’s disease-like phenotype [17]. Although the carrier frequency of the variants remains low, we believe that this new kit, based on the genotype–auditory correlation, could be used widely for specific auditory phenotypes, as a part of precision medicine. Likewise, we will continue to update new key genetic variants for deafness, which will continuously be revealed through NGS, as targets for the newly developed genotyping kit. The clinical applicability of this phenotype-specific and variant-specific genotyping kit surely outpowers that of NGS in a routine everyday clinical setting, especially when we are aware of the phenotype and can assume candidate genes.

## 5. Conclusions

Taken together, our results suggest that U-TOP™ HL Genotyping Kit Ver2 for the screening of 11 variants from five deafness genes, comprised of two ANSD genes (*OTOF* and *ATP1A3*) and three SNHL genes (*MPZL2*, *COCH*, and *TMC1*), can be used as a reliable screening tool in the Korean population with a distinctive auditory phenotype. Particularly, this newly developed kit is highly specialized for diagnosing prelingual ANSD, allowing more efficient, rapid, and cost-effective etiologic diagnosis. Therefore, we have provided a clinical guideline on the management of prelingual ANSD children in this era of precise medicine and customized auditory rehabilitation.

## Figures and Tables

**Figure 1 diagnostics-10-00672-f001:**
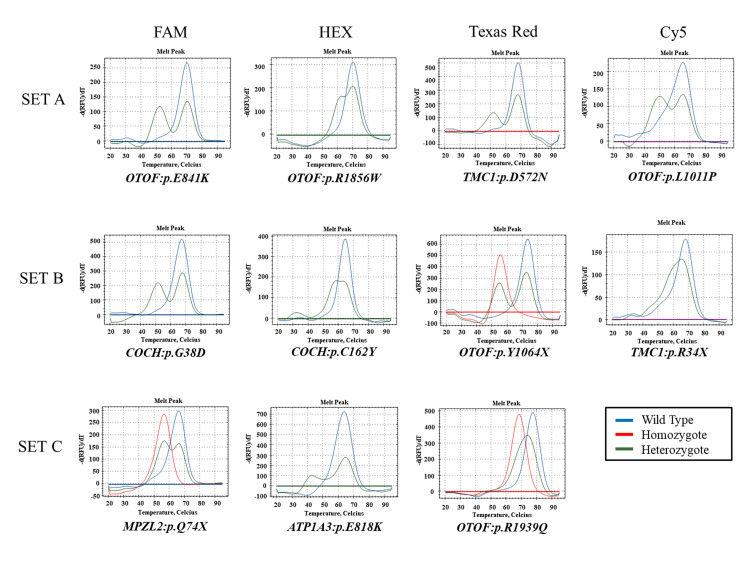
Melting peaks of 11 variants from five deafness genes comprising two ANSD genes (*OTOF* and *ATP1A3*) and three SNHL genes (*MPZL2*, *COCH*, and *TMC1*) with a distinct auditory phenotype among Koreans. The blue, green, and red lines indicate wild type, heterozygous, and homozygous mutants, respectively. Each hearing loss set contains 3 or 4 peptide nucleic acid (PNA) probes labeled with FAM, HEX, Texas Red, and Cy5 channels.

**Figure 2 diagnostics-10-00672-f002:**
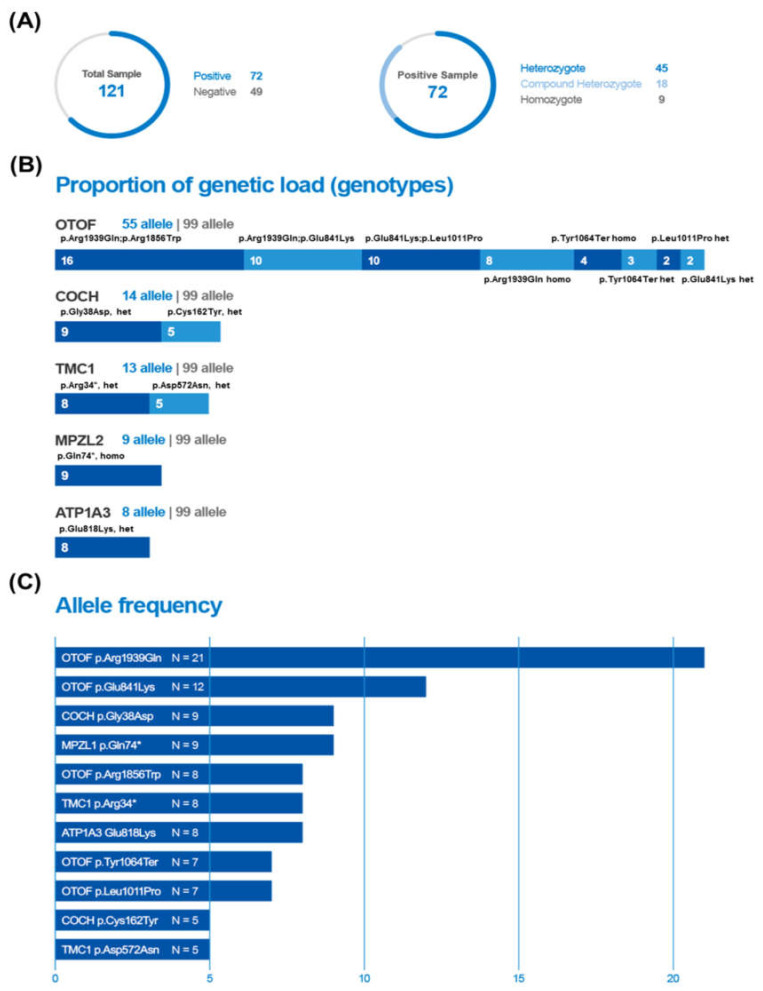
Characteristics of samples for validation of the efficacy of the U-TOP™ HL Genotyping Kit Ver2. (**A**) Composition of a total of 121 samples (72 samples with variants and 49 negative control samples). Positive samples (*n* = 72) carried at least one of the 11 target variants as a single heterozygous (*n* = 45, 62.5%), compound heterozygous (*n* = 18, 25.0%) or homozygous (*n* = 9, 12.5%) state. (**B**) Ninety-nine alleles from 72 positive samples and 15 different genotypes. The proportion of genetic load was depicted in the order of frequency of genotype. For example, the proportion of genetic load of *OTOF* among the 99 alleles was 55.6%, with 8 different genotypes. (**C**) Allele frequency according to each variant. The variant p.Arg1939Gln of *OTOF* was the most common, with an allele frequency of 22.1%.

**Figure 3 diagnostics-10-00672-f003:**
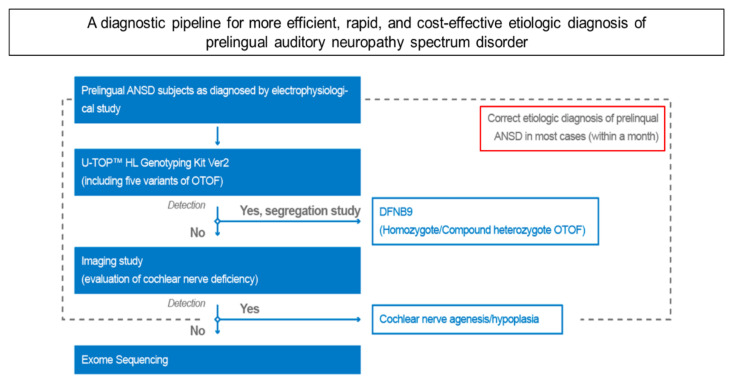
A diagnostic pipeline for more efficient, rapid, and cost-effective etiologic diagnosis of prelingual auditory neuropathy spectrum disorder (ANSD). This diagnostic strategy involves screening of the five most prevalent *OTOF* variants using a newly developed genotyping kit, imaging study, and exome sequencing for correct etiologic diagnosis of prelingual ANSD.

**Table 1 diagnostics-10-00672-t001:** Comparison of results between U-TOP™ HL Genotyping Kit Ver2 and Sanger sequencing.

		Sanger Sequencing		Total
		Positive	Negative	
**U-TOP™ HL Genotyping Kit Ver2**	Positive	72	0	72
	Negative	0	49	49
Total		72	49	121

**Table 2 diagnostics-10-00672-t002:** Phenotypic spectrum and etiologic diagnosis of six consecutive additional subjects with prelingual auditory neuropathy spectrum disorder (ANSD).

Family ID	Age (Months)	Sex	Audiological Assessment			U-TOP™ HL Genotyping Kit Ver2	IAC MRI Analysis	Elapsed Time for Diagnosis ^a^ (Days)
			ABR (dB)	CM	DPOAE	Genotype Variant (*OTOF*) NM_001287489		
SB543-1015	5	F	(B) NR	(B) response	(B) response	c.5816G>A:p. Arg1939Gln (Homo)	NA	11
SB539-1011	7	F	(B) NR	(B) response	(R) partial (L) NR	Not detected	(B) CND	25
SB515-981	11	F	(B) NR	(B) response	(B) NR	Not detected	Normal	27 ^b^
SH336-742	38	M	(B) NR	(B) response	(B) response	c.5816G>A: p. Arg1939Gln (Homo)	NA	15
SB527-1026	6	F	(B) NR	(B) response	(R) response (L) partial	Not detected	(B) CND	52
SB554-1029	5	F	(B) 70	NA	(B) response	Not detected	Normal	14 ^c^

Abbreviation: ANSD, auditory neuropathy spectrum disorder; ABR, auditory brainstem response; CM, cochlear microphonics; DPOAE, distortion-product otoacoustic emissions; IAC MRI, internal acoustic canal magnetic resonance imaging; mo, months; F, female; M, male; B, both; NR, no response; Homo, homozygote. ^a^ Note that this is time to correct etiology of prelingual ANSD of these six subjects from the day of the initial visit. ^b^ Note that the etiology of prelingual ANSD for SB515-981 turned out to have predisposing factors associated with ANSD, such as prematurity and hypoxic damage. ^c^ Note that the hearing of SB554-1029 in whom neither the genotyping kit nor IAC-MRI revealed any abnormality, improved over time.

## Data Availability

The datasets generated during and/or analyzed during the current study are available from the corresponding author on reasonable request.

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
