# Peer review of "Flexible Real-Time Polymerase Chain Reaction-Based Platforms for Detecting Deafness Mutations in Koreans: A Proposed Guideline for the Etiologic Diagnosis of Auditory Neuropathy Spectrum Disorder"

_diagnostics, 2020, doi:10.3390/diagnostics10090672_

Round 1

Reviewer 1 Report

Lee and colleagues describe the development of a real-time PCR-based kit (U-TOP™ HL Genotyping Kit Ver2) for 11 recurrent variants related to auditory neuropathy spectrum disorder (ANSD). The group previously reported the development of a genotyping approach, using the same technology, designed to identify high prevalent variants in 5 genes related to pre-lingual deafness in the Korean population.

The authors highlight the relevance of testing for high-frequency variants using relatively cheap and rapid approaches, rather than beginning with next generation sequencing (NGS). As such, the central theme of the work is important and has clinical relevance. If this work were to be published, I would expect the following major issues to be addressed.

  1. The justification for the variants chosen in this panel should be presented more clearly. The authors highlight that the variants were selected “based on previous Korean and East Asian reports from several leading institutes, including ours…” This data should be presented more clearly within the manuscript and could benefit from a table.
  2. The U-TOP approach needs to be explained in more detail earlier within the manuscript.
  3. On line 131 (Section 2.3) the authors state that McNemar’s test was used to show that the sensitivity and specificity of their assay was 100%. McNemar’s test used to determine if there are differences on a dichotomous dependent variable between two related groups. It is not used to establish sensitivity and specificity. The authors need to clarity their use of this statistical tool as, currently, it seems to be being used inappropriately.
  4. On line 134 the authors state that “Compared with the reference values of Sanger sequencing (94% sensitivity and 92% specificity), the newly developed genotyping kit showed significantly higher sensitivity and specificity.” It is unclear where these sanger values have been chosen from (presumably the literature) and a few lines later the authors describe the validation of their kit against Sanger with 100% concordance. As such, stating that their approach is superior to sanger sequencing is unjustified and confusing. This section requires significant rewording.
  5. The equation in section 2.3 describing the mathematical basis for Cohen’s adds little and should be removed.
  6. On line 150 the authors go onto describe the clinical implications of their kit. They undertook genotyping on 6 prospectively recruited, consecutive cases with pre-lingual ANSD. 2 of the patients had homozygous variants in Although interesting, this small cohort is not sufficient to label the diagnostic yield as 33%, as the authors do. Ideally, a longer validation period should be undertaken or, if not possible, the authors should highlight this as a significant limitation.
  7. The work would benefit from a cost-benefit analysis – detailing the economic benefit of testing patients using a SNP genotyping approach prior to NGS.

Reviewer 2 Report

Well written and very interesting paper on the diagnostic possibilities of the ANSD. The only thing that I do not undestand is why you put the Material and Methods Section after the Results and Discussion and not in the typical sequence.

Round 2

Reviewer 1 Report

The authors have addressed the comments I raised in my initial review.